# Bacteriophages in Infectious Diseases and Beyond—A Narrative Review

**DOI:** 10.3390/antibiotics12061012

**Published:** 2023-06-05

**Authors:** Petros Ioannou, Stella Baliou, George Samonis

**Affiliations:** 1School of Medicine, University of Crete, 71003 Heraklion, Greece; 2Internal Medicine Department, University Hospital of Heraklion, 71110 Heraklion, Greece

**Keywords:** bacteriophage, antimicrobial resistance, multi-drug-resistant, extensively drug-resistant, pan-drug-resistant

## Abstract

The discovery of antibiotics has revolutionized medicine and has changed medical practice, enabling successful fighting of infection. However, quickly after the start of the antibiotic era, therapeutics for infectious diseases started having limitations due to the development of antimicrobial resistance. Since the antibiotic pipeline has largely slowed down, with few new compounds being produced in the last decades and with most of them belonging to already-existing classes, the discovery of new ways to treat pathogens that are resistant to antibiotics is becoming an urgent need. To that end, bacteriophages (phages), which are already used in some countries in agriculture, aquaculture, food safety, and wastewater plant treatments, could be also used in clinical practice against bacterial pathogens. Their discovery one century ago was followed by some clinical studies that showed optimistic results that were limited, however, by some notable obstacles. However, the rise of antibiotics during the next decades left phage research in an inactive status. In the last decades, new studies on phages have shown encouraging results in animals. Hence, further studies in humans are needed to confirm their potential for effective and safe treatment in cases where there are few or no other viable therapeutic options. This study reviews the biology and applications of phages for medical and non-medical uses in a narrative manner.

## 1. Introduction

In the fight against infectious diseases, the application of hand hygiene and the discovery of antibiotics are the two most important innovations that have radically changed the course of history through the reduction of infection incidence as well as through the reduction of morbidity and mortality associated with infectious diseases. However, right after the discovery of penicillin by Fleming and the widespread use of antibiotics for medical purposes in the 1940s, it became quite evident that inappropriate use of these agents could lead to the development of resistant microorganisms [1,2]. This phenomenon quickly became present, thus mitigating the enthusiasm that followed the discovery of antibiotics [3]. Interestingly, though, evidence suggests that antimicrobial resistance preceded the medical use of antibiotics, as in the case of penicillin, where bacterial penicillinases seem to have been detected before the use of this agent, with an important number of antibiotic genes being present in natural microbial populations [4,5].

For many decades, antibiotic development surpassed antibiotic resistance. Hence, the problem remained dormant. However, in the last decades, the antibiotic pipeline slowed down [6]. This was accompanied by an increase in resistance of many pathogens, often leading to the development of infections with few therapeutic options, as in the case of multi-drug-resistant (MDR), extensively drug-resistant (XDR), and pan-drug-resistant (PDR) pathogens [7]. Infections by such pathogens are associated with increased mortality, while therapeutic options often include the revival of older antibiotics such as polymyxins or antibiotic combinations [8,9,10,11]. For example, *Acinetobacter baumannii* is a microorganism classically involved in hospital-acquired infections that has gained considerable antimicrobial resistance during the last decades, and this has left few therapeutic options, namely colistin. On the other hand, the development of PDR strains left no clear therapeutic option, leading to the use of antimicrobial combinations of two or three drugs, usually including colistin, tigecycline, or other antimicrobials, in a method of bending the mechanisms that render this pathogen resistant [12,13,14]. Importantly, even though many substances enter the antibiotic pipeline to be tested in clinical studies and be approved for clinical use, most of them fail during evaluation and are, therefore, rejected [15]. Of note, even though there are still new agents approved for clinical use, they mostly belong to specific, already-existing classes, meaning that there is a lack of new antibiotic categories and novel treatments for problematic pathogens such as MDR, XDR, and PDR [16]. Hence, during the last decades, there were only three reported new antimicrobial classes identified [17,18,19]. Due to the molecular similarity between the different antimicrobials that belong to the same class, as a result of synthetic modifications to pre-existing agents, the development of cross-resistance or the evolution of even broader resistance mechanisms, such as extended spectrum beta-lactamases, is a relatively frequent phenomenon, quickly reducing the use of these compounds [20].

Based on these data, it becomes clear that there is an imperative need for the development of new approaches either in terms of classical antimicrobials or in terms of novel compounds such as antimicrobial peptides or bacteriophages that could be used in the fight against infectious diseases and, more specifically, against drug-resistant microorganisms. The aim of the present study is to review the history, the present condition, and the future implications of phage therapy against infectious diseases.

## 2. Definitions and Biology of Phages

Bacteriophages (phages) are viruses with a size of 20 to 200 nm that infect bacteria with very high specificity [21]. For example, a phage not only targets particularly a specific species within a genus, but it usually targets a very specific subset of bacterial strains within the species. Hence, in an effort to identify potential phages for the treatment of a young boy with cystic fibrosis infected by *Mycobacterium abscessus*, a collection of more than 10,000 phages that were isolated using *Mycobacterium smegmatis* needed to be screened to identify just three useful phages [22]. In the middle of the previous century, bacteria and phages were used in experiments elucidating basic questions in biology and genetics [23,24,25]. Due to their bactericidal activity and their inability to infect eukaryotic cells, phages could be used in the fight against infectious diseases. The idea is not new, but the development of this kind of therapy was restrained by inadequately controlled trials and the discovery of new antibiotics. However, nowadays, the development of phage therapy may be facilitated by easier methods of bioengineering. Furthermore, phages’ biodiversity in nature could be used as an advantage for the identification and development of phages that are ideal for fighting specific bacteria [26,27,28,29].

Phages were discovered independently by two scientists: the British bacteriologist Frederick William Twort in 1915 and the French-Canadian microbiologist Felix d’Herelle in 1917. D’Herelle noticed that these entities had the ability to kill bacteria and named them “bacteriophages” [30]. Phages are similar to other viruses in that they have a nucleic acid genome that is enclosed in a protein capsid that protects the genetic material and plays a role in its delivery into the target bacterial cell. Most phages have complex morphologies, far beyond the classical helicoidal or icosahedral morphologies encountered in the case of other viruses [31]. Most phages belong to the order *Caudovirales*, which are tailed viruses (cauda is the Latin word for “tail”) and, according to the Baltimore classification scheme, are group I viruses with double-stranded DNA (dsDNA) for their genome, with a length of 3400 to 500,000 base pairs [32]. Other types of phages include helical, isometric, and pleomorphic viruses, even though they comprise a minority. Isometric phages may have any type of genome, whereas pleomorphic and helical phages usually contain dsDNA genomes [31].

The life cycle of a phage involves its binding to a receptor of a bacterium that leads to releasing of the genetic content of the phage into the bacterium. This binding occurs after the recognition of a receptor of the bacterium that allows the phage to attach to and enter the bacterial cell. These receptors are commonly proteins or sugars on bacterial cells recognized by proteins of phages that mediate their adhesion on the bacterial cell, leading to the entry into the bacterial cell [27]. An example is the binding of phage T4 to susceptible *Escherichia coli* through two sets of tail fibers [33]. Other examples of a phage binding to receptors include the tail spike of phage SPP1 used to attach *Bacillus subtilis* or the baseplate proteins used by some *Siphoviridae* phages to initiate the infection of *Lactococcus lactis* [34,35]. Several other examples of a phage binding to bacterial cells have been described, with significant variation in the biology of the interaction of phage proteins with bacterial receptors. However, the biology of these interactions and the biodiversity imply that there is much yet to be discovered [27,36]. Until now, efforts to use phages as treatment of infectious diseases did not always take into consideration the phage binding to the bacterium [27]. Current efforts to use phages as therapeutic tools should include studying the binding site of the phage as well as the possibility of adaptive changes in the bacterial receptors that may reduce the binding procedure and thus cause resistance to the phage used [27].

After the binding of the phage to the bacterial receptor and the genome injection into the cell, the genetic material of the phage takes advantage of the molecular machinery of the host bacterium and undergoes transcription, translation, and replication, leading to the formation of the required components of new phages that are, further on, released with concomitant lysis of the bacterium. Then, this process is further repeated by the new viral particles after the infection of new bacterial cells. Hence, using phages as an antibacterial treatment has the theoretical advantage of using a self-amplifying therapeutic modality, contrary to the therapeutic mode of classical antibiotics that require repeated doses and, often, prolonged treatment for achieving the required clinical result [27]. However, there are two different types of phages in terms of proliferation that are of clinical significance. One type is the lytic phage, which always follows the previously mentioned cycle upon infection of a bacterium, leading to bacterial destruction and concomitant phage proliferation. The other type of phages is the lysogenic ones, which integrate their genetic material into the bacterial genome and do not necessarily lead to bacterial lysis right after they infect the bacterial cell. Then, the phage genome is inherited by the daughter bacterial cells through binary fission, and upon specific environmental or physiological stressors that negatively affect the fitness of the bacterial host cell, the phage starts a lytic cycle, as previously described [27]. Lytic phages seem more appropriate for clinical use, as they are anticipated to more readily cause bacterial damage at a higher extent, while they also have the theoretical benefit of being safer compared to lysogenic phages since they do not involve gene transfer. Additionally, lysogenic phages could be of use in other settings, such as the monitoring of tumors as well as in several fields of biomedical research [37,38,39]. Figure 1 and Figure 2 summarize the life cycles of a lytic and a lysogenic phage, respectively.

## 3. Non-Medical Applications of Phages

### 3.1. Applications on Food Safety

Phages have been long used for prophylaxis and treatment in cattle, where the most common infectious diseases are mastitis, metritis, or respiratory tract infections, and most of them are due to bacterial causes [40]. For a long time, antimicrobial substances, either classical antibiotics or antimicrobial peptides, have been used in this field. However, their extensive use has raised environmental concerns as well as concerns regarding the effects on human health. To that end, studies focusing on bacterial causes of mastitis in farm animals have shown that the use of phages or lytic proteins deriving from phages could be of use to overcome even resistant bacteria such as *Staphylococcus aureus* or *Escherichia coli* [40,41,42,43].

Foodborne illnesses result in hundreds of millions of human infections and hundreds of thousands of deaths every year, posing a significant public health issue [44,45]. Decontamination of foods is challenging, as the strategies used may be either inadequate or corrosive and may damage the food [31]. Hence, treatment of food with water may not have any significant effect on microorganism load, while treatment with chemicals may decrease the number of microorganisms but may also cause chemical changes to food proteins, while some microorganisms may develop resistance to them. To that end, the use of lytic phages in this context may be of use, leading to a significant reduction of pathogenic bacteria capable of causing foodborne disease [46,47,48]. Furthermore, the inability of phages to infect eukaryotic cells implies that they may not infect humans, while they are also not anticipated to alter the food [49,50].

As shown by several studies, the direct application of lytic phages to food that is ready to be used can reduce the number of potentially hazardous foodborne bacteria in a significant manner [47,51,52,53]. For example, when lytic phages were applied to chicken skin that had been previously contaminated with *Campylobacter jejuni* or *Salmonella enterica* serovar Enteritidis, the phage titers increased, and the pathogen bacterial load was reduced [51]. In another study, a virulent phage cocktail targeting *Salmonella flexneri*, *Salmonella dysenteriae*, and *Salmonella sonnei* was found to effectively reduce the likelihood of contamination of ready-to-eat chicken products by *Shigella* spp. [53]. These data have led to the approval of several phages for use in food decontamination [31]. The U.S. Food and Drug Administration (FDA) approved a mixture of six phages (ListShield, Intralytix Inc., Columbia, IN, USA) as a food additive for ready-to-eat poultry and meat products in order to control contaminations caused by *Listeria monocytogenes* [54]. Other similar products such as EcoShield (Intralytix Inc., Columbia, IN, USA), which includes three lytic phages targeting *E. coli* O157:H7; SALMONELEX (Micreos Food Safety, Wageningen, The Netherlands) targeting *Salmonella* spp.; and Listex P100 (Micreos Food Safety, Wageningen, The Netherlands), which consists of a single phage targeting *L. monocytogenes*, are currently available [31].

Moreover, phages can be of great help in food safety if used in biosensors to detect pathogens that may contaminate food. For example, current methods for the diagnosis and analysis of food samples are more laborious and time-consuming than the application of biosensors that could occupy specific phages to detect food contamination by specific bacteria. These could take the form of biosensors that could use bio-probes and transducers in the biosensor to allow timely identification of food contamination [55,56].

Another aspect regarding food safety has to do with biofilms. Biofilms are complex structures that are hard to remove or eradicate with the use of disinfectants, and they have inherently very high resistance to antimicrobials, as they have several layers of microorganisms with different levels of metabolic dormancy as well as a rich extracellular matrix that also poses a barrier for the diffusion of antimicrobials. Biofilms pose a threat to consumers of dairy products. Phages could also be of use for decontamination of inanimate surfaces that may host biofilms, thus reducing the likelihood of disease transmission across the dairy chain [31,40,57]. Indeed, there is evidence that lytic phages can reduce biofilms on several surfaces, including those of pathogenic bacteria that are causes of foodborne diseases, such as *E. coli* O157:H7, *L. monocytogenes*, or *Salmonella* [47,58,59,60,61]. There are already products such as ListexP100 and Listshield (Intralytix Inc.) that are indicated for the reduction of the levels of *L. monocytogenes* on non-food equipment, as they prevent the formation of biofilm and may even help to eliminate it if it has been previously formed [62]. However, even though the use of phages may sound promising in this context, and it does indeed seem to be a viable option against biofilm in industrial settings, it does have some disadvantages, such as issues regarding large-scale production or the high specificity of phages for specific bacteria, which means that most of the microorganisms that may contaminate surfaces will not be affected. Thus, the use of phages is not anticipated to replace that of disinfectants, but it could, in specific circumstances, act in a complementary way [31,63]. The regulatory status of phage use is not standardized globally. Some factors that can influence the regulatory landscape regarding the use of phages in food safety include the following:(a)Novel Food Regulations: Many countries have regulations governing the approval and use of novel food ingredients, which can include phages. These regulations are designed to ensure the safety and proper labeling of new or non-traditional foods. Phages used in food applications may need to undergo a regulatory approval process to demonstrate their safety and efficacy before being permitted for use;(b)Risk Assessment and Safety Evaluation: Regulatory authorities typically require a thorough risk assessment and safety evaluation for novel food ingredients, including phages. This evaluation may include determining the potential for adverse effects on human health, assessing the likelihood of gene transfer or antibiotic resistance development, and evaluating the stability and persistence of the bacteriophage in the food environment;(c)Codex Alimentarius: The Codex Alimentarius Commission is an international body that develops food standards, guidelines, and codes of practice. Codex standards provide a reference for national regulatory authorities when developing their own regulations. The use of phages in food safety may be subject to Codex guidelines or specific regulations implemented by individual countries in line with Codex recommendations;(d)Labeling and Consumer Information: Proper labeling and consumer information are important aspects of food regulations. Regulatory authorities may require clear and accurate labeling of foods treated with phages, including information on the presence of phages, their specific targets, and any necessary handling or storage instructions;(e)Country-Specific Regulations: Each country has its own regulatory framework for food safety, including the use of novel ingredients such as phages. The requirements and approval processes can differ significantly between countries due to variations in risk assessment methodologies, regulatory structures, and levels of acceptance for novel technologies [46,64,65,66,67,68,69].

### 3.2. Applications in Agriculture

Some plant pathogens may cause disease leading to severe financial losses in agriculture by affecting the quality of products and reducing the production yield. Phages could be an option in agriculture for allowing the reduction of plant pathogens and reducing their hazardous effects [70]. Indeed, several studies have proven the efficacy of phages against plant diseases, thus improving the quality and yield of agricultural production [31,71,72,73,74]. For example, by using phages shown to have activity against the pathogen *Ralstonia solanacearum*, a phage cocktail, P1, that contained six phages was used in preventing potato bacterial wilt by decontamination of sterilized soil spiked with *R. solanacearum* or by injecting the phage cocktail into the plants [74]. In another example, phage PE204 was used as a lytic phage to investigate its ability to control bacterial wilt on tomato plants. Concomitant treatment of phage PE204 with *R. solanacearum* on a tomato rhizosphere led to complete inhibition of bacterial wilt occurrence, while amendment of the surfactant Silwet L-77 at 0.1% to the phage suspension did not affect its disease control activity [73]. Currently, there are commercially available options of phages for use in agriculture, such as Agriphage (Omnilytics Inc., Sandy, UT, USA), which contains phages against *Xanthomonas campestris* pathovar *vesicatoria* and *Pseudomonas syringae* that are responsible for causing disease in tomatoes and peppers, or AgriPhage-Fire Blight (Omnylitics Inc, Sandy, UT, USA, Columbia, IN, USA), which was approved for use against fire blight (caused by *Erwinia amylovara*) in pears and apples [31]. These products were found to increase yields and reduce the likelihood of bacterial spots compared to copper compounds [31].

### 3.3. Applications in Aquaculture

Some species of the genera *Lactococcus*, *Vibrio*, *Pseudomonas*, and *Aeromonas* have the potential to cause disease in cultured fish and shellfish, while they could also, under specific circumstances, cause disease in humans as well [75]. To that end, the use of phages in aquaculture could lead to the control of bacterial disease and the optimization of industrial production [72,76]. Several studies have shown the potential of phages to control bacterial diseases in aquaculture, even those caused by resistant pathogens, implying that phages could be an alternative to antibiotics in this setting [77,78,79,80,81,82,83,84]. For example, in a recent study, the *Edwardsiella tarda* phage (ETP-1) that was isolated from marine fish-farm water was found able to control infection by the pathogenic MDR *E. tarda* in zebrafish aquaculture [83].

### 3.4. Applications in Wastewater Plant Treatment

Contamination of wastewater plants by waterborne bacteria is of global concern due to the associated morbidity and mortality as well as the high costs required for these plants’ disinfection. Some bacteria such as *Vibrio*, *E. coli*, *Shigella*, and *Salmonella* are of particular concern for human health, and their control in this setting is a priority. Hence, the likelihood of human disease can be reduced without affecting antimicrobial resistance [85]. In particular, phages could be of use both for wastewater plant treatment and as indicators due to their specificity for bacteria, thus acting as tracers of pathogens for monitoring wastewaters [31,86]. However, there are some concerns regarding the use of phages in these settings. For example, there are important differences in the microbiology of different plants, while very large numbers of phages are required to control bacterial pathogens [86,87].

### 3.5. Applications as Hospital Environment Sanitizers

Hospital-acquired infections (HAIs) are a leading cause of morbidity and mortality in hospitals and are closely associated with antimicrobial resistance [88,89,90,91,92]. Surfaces in hospitals are usually persistently contaminated by several bacteria representing a serious threat due to their potential transmission and colonization of patients. These microorganisms, under specific circumstances, may lead to HAIs [93,94,95,96]. Such organisms include *S. aureus* and, more specifically, methicillin-resistant (MRSA) strains that are of particular concern due to difficulties in treatment as well as *Pseudomonas aeruginosa* and *E. coli*, which are among the most common causes of HAIs and are also associated with particularly problematic antimicrobial resistance patterns [97,98,99,100]. The cleaning of hospital surfaces usually involves the use of several disinfectants. However, there are issues that have to do with the applicability, implementation, and efficacy of the currently available disinfectants that may render these options partially ineffective, at least for some specific pathogens [101,102,103]. Furthermore, the theoretical concern about the development of resistance to chemicals as well as to antibiotics, along with the fact that disinfectants do not eliminate only the pathogenic bacteria but also those that could be protective against colonization by resistant hospital microorganisms, strengthens the need to identify additional measures for sanitizing hospital surfaces [31,104,105].

There are studies assessing the effect of phages against pathogens that could be isolated from hospital surfaces, such as *E. coli*, *Salmonella*, MRSA, or *A. baumannii*, and their results are indeed promising [47,61,106,107]. However, applicability in hospital settings in everyday practice may have some caveats since experimental conditions in studies may differ significantly from those in hospital settings. In studies, phages were used against very high densities of bacteria, allowing optimal interaction between them and the bacteria, while the interaction was mediated by the application of an aqueous solution containing the phages that were applied on surfaces for a long time; these are conditions that cannot be fulfilled easily in everyday hospital settings [47,106]. Interestingly, however, there are studies assessing the use of phages in hospitals that showed promising results. Hence, in a study using phages for the decontamination of an intensive care unit (ICU) and specifically targeting carbapenem-resistant *A. baumannii* along with standard cleaning procedures, the occurrence of HAIs due to this microorganism was substantially reduced, implying that phage decontamination could be of practical use in the fight against HAIs [108].

In an effort to promote clever and eco-friendly cleaning by respecting non-pathogenic bacteria on hospital surfaces, phages were evaluated in addition to eco-friendly detergents (probiotic cleaning hygiene system (PCHS)) containing non-pathogenic probiotic bacteria of the *Bacillus* genus in an effort to control the microbiome of the hospital [109,110]. To that end, the addition of phages targeting pathogenic bacteria could amplify the decontamination of hospitals with respect to bacteria that could be used as a barrier against pathogenic organisms, while, importantly, the phages added have no activity against the non-pathogenic probiotic bacteria contained in the PCHS. This probiotic cleaning system has already been shown to be effective and safe for patients, leading to a gradual and stable change in the microbiology of hospital surfaces, including the reduction of pathogens with increased antimicrobial resistance, leading to a significant decrease in the acquisition of HAIs, antimicrobial use, and hospital costs [105,111,112,113,114]. Hence, the use of phages in the context of PCHS could achieve faster cleaning and/or eradication of specific pathogens that have colonized specific hospital areas or wards, as is the case of the XDR and PDR pathogens prevailing in ICUs [31,90,115]. In vitro tests have shown that if PCHS are combined with phages, they retain their stability and activity, allowing them to target bacteria with significant antimicrobial resistance that are often also resistant to disinfectants, thus leading to an additive or synergistic effect [104,109,116]. A study evaluating the effect of the combination of PCHS and phages against *Staphylococcus* species in bathrooms, which are the most heavily contaminated areas of hospitals, showed that daily sanitation by the combination led to a rapid and significant decrease in the levels of *Staphylococcus* spp. on the surfaces where it was applied, and it was 97% more effective as compared to the use of PCHS alone [110].

Notably, phages can be used not only for the disinfection of hospital surfaces and medical equipment from free bacteria but could also be of use for disinfection from biofilms [117]. For example, a study on the effect of phage use and chemical disinfection against *P. aeruginosa* showed that phages can be successfully combined with chemical disinfectants such as benzalkonium chloride and sodium hypochlorite to increase the efficacy of wet biofilm and bacterial spot removal on surfaces and also reduce the likelihood of biofilm regeneration [118].

Moreover, phages could be also used in medical applications in non-human settings if used in biosensors, as in food safety, since in that case biosensors could use bio-probes and transducers to allow timely bacterial contamination in critical specimens and surfaces [55,56]. Figure 3 summarizes the non-medical applications of phages.

## 4. Medical Applications of Phages

### 4.1. Early Reports of Medical Use and Drawbacks

Immediately after the discovery of phages by Twort and d’Herelle, d’Herelle postulated that phages could have therapeutic applications. In 1919, he used phages to treat chickens that were infected by *Salmonealla gallinarum* with success [119,120]. Based on the successful use in animals, d’Herelle realized that phage treatment could be attempted in humans with bacterial disease. In 1921, five patients suffering from bacillary dysentery were treated successfully with the use of a phage targeting *Shigella dysenteriae* [119,121]. Furthermore, clinical trials of phage treatment in cholera in India showed a significant decrease in mortality from 62.8% in the control group to 8.1% in the group treated with phages, while d’Herelle noticed that when adding phages targeting cholera into drinking wells during outbreaks of the disease, a reduction of subsequent infections was noted [122].

After these encouraging results in animals and humans, other scientists recognized phages’ potential in prophylaxis and treatment and started targeting other infections, although with partial success. Beyond criticism regarding the quality and the design of these studies in humans, an important drawback at that time was the phages’ high specificity against certain bacteria, implying that previous recognition of the pathogen was absolutely necessary to allow successful treatment by phages [27]. In 1923, phages were successfully used for the treatment of bacteremic patients with typhoid fever. However, at that time, other scientists reportedly failed to achieve the same results using a similar patient population [123,124]. This inconsistency was considered to be due to the use of a phage with a very narrow spectrum. Other drawbacks included issues regarding phage production, involving contamination of the product due to imperfect filtering and purification steps. Additionally, pharmacokinetics suggested rapid phage elimination from patients’ spleen, rendering the therapeutic effect short-lived. Furthermore, the bacterial ability to rapidly mutate and develop resistance to phages probably played an important role. More specifically, bacteria have multiple antiviral mechanisms that may inhibit the entry of a phage (by blocking their interaction with the bacterial receptors or producing an extracellular matrix that may reduce the likelihood of the interaction leading to phage entry in the bacterial cell or may produce competitive inhibitors that may antagonize phage binding on the bacterial cells), may prevent DNA entry into the bacterial cell, may inactivate phage DNA even after entry, or may lead to abortion of the infection by even inducing bacterial cell self-death [125]. Due to the adaptive nature of these mechanisms, there is a possibility for the development of resistant bacterial clones during phage treatment [126]. Finally, the possibility of a lag between in vitro and in vivo experiments cannot be excluded [23,27,127,128,129,130]. Thus, given these issues and the fact that the interest in the production of antibiotics was on the rise, interest in phage therapy waned for many decades in the West but notably persisted in the USSR and Eastern Europe. Nowadays, with the problem of the increasing antimicrobial resistance that threatens global health and sets at risk millions of hospitalized patients, interest in phage therapy is on the rise again in Western countries [27].

### 4.2. Rediscovering Phage Therapy—The 1980s

A series of well-designed experiments by Smith and Huggins helped to rediscover phage therapy, as they assessed several issues that had been identified as limitations in the previous sets of experiments in the first half of the 20th century and also confirmed that phage treatment is safe and efficient in animal models [27]. In the first series of experiments, Smith and Huggins showed that there was a close in vitro to in vivo correlation of phage efficacy. For experiments, they chose phage R that targets K1+ *E. coli*, showing the greatest results in in vitro virulence [131]. In the following experiments, they showed that a single dose of phage R was equally effective as eight doses of streptomycin [131]. Furthermore, bacterial lysate, free of phages, had no therapeutic effect in the infected animals. Intramuscular injection of phages into uninfected mice proved that they persisted inside the injected muscle as well as the spleen for four weeks after injection. However, pathogenic bacteria were cleared within 16 and 20 h after phage injection from the liver and the blood, respectively. Finally, bacterial mutants resistant to R phages were K1-, occurring at a frequency of about 0.01, while they were known to be avirulent. With this study, most of the concerns described previously were addressed, and phage treatment was presented as at least equal to or even more effective than antibiotics [27].

In another set of experiments, Smith and Huggins investigated the issue of bacterial resistance to phages, choosing an *E. coli* diarrhea model in calves. They developed a multi-phage treatment plan to combat the development of resistance during treatment. Before experimenting in vivo, they performed in vitro studies. Hence, they chose lytic phage B44/1, which had the ability to infect only K85+ strains of *E. coli*. Then, they isolated bacterial mutants that were resistant to that phage in vitro. Subsequently, they chose another phage, namely B44/3, which was able to infect bacteria resistant to the initial B44/1 phage that they used. Additionally, they selected bacterial mutants resistant to B44/3 but susceptible to B44/1. By evaluating the dynamics of phage resistance before using them in vivo, they were able to use this double-phage approach to overcome the problem of resistance evolution to phages [132]. Smith and Huggins also evaluated phage stability during treatment given orally. They noticed that although there was poor phage stability in the stomach due to an acidic environment, this could be overcome by administering calcium carbonate before the oral administration of phage [133]. Thus, this group of scientists undermined all that was previously considered as significant drawbacks for phage treatment, opening the way for further experiments and giving hope for phage application [27].

New experiments from other research groups have further elucidated questions on phage biology and therapeutics. A more recent study showed that even though rapid clearance of phages in the blood was considered a drawback, it is possible to select phage variants that have longer blood half-lives [134]. Other experiments showed that phage treatment was also efficacious against pathogens such as *P. aeruginosa* or *A. baumannii* [135]. Moreover, in another set of experiments, a phage targeting the K1+ *E. coli* provided 100% protection to mice, while treatment with phage not targeting K1 led to a mortality of 60%. A comparison with streptomycin revealed that the phage targeting K1 led to 9% mortality, while treatment with streptomycin was associated with 54% mortality, thus confirming the previous findings by Smith and Huggins [136].

### 4.3. The Current Era

The 21st century offers many more tools than the previous one, such as high-throughput methods for efficiently screening thousands of samples at the same time, affordable whole-genome sequencing, automated technology for microbiological techniques, etc. On the other hand, clinical studies have shifted to a different level, with a need for more adequate design for power, double-blind and randomized settings, as well as higher standards regarding safety for participants. Current technology allows deeper investigations at the time of the study or even after that, addressing clinical and biological questions that may have to do with the in vitro as well as the in vivo interaction of the phage with the target bacterium and the immune system of the human host [27].

Importantly, the well-known physiological property of very narrow targeting of phages that could be considered a limitation in some instances could be addressed with the current technology. The work by Dedrick et al. showed an example of how narrow the targeting of phages is and how phage engineering could be elaborated. In this study, a collection of more than 10,000 phages isolated using *Mycobacterium smegmatis* was needed for screening to allow the identification of just three useful phages that were used for the treatment of a young boy with cystic fibrosis and infection by *Mycobacterium abscessus*. Two of them were engineered to allow appropriate bacterial targeting [22]. Nowadays, CRISP-Cas9 technology can also be used to engineer the genomes of phages to manipulate their bacterial targeting [137,138].

Regarding the development of resistance to phages by bacteria, modern technologies of evolutionary biology could provide insights about appropriate phage selection. In particular, the selection of a phage targeting a specific molecule as a receptor leads to the selection of bacterial strains that do not express it. Hence, therapeutic use of phages could take advantage of the receptors that are concomitantly virulence factors. For example, the selection of strains that do not produce the virulence factor that is the target receptor for the phage might lead to bacterial survival; however, their virulence would be lower, leading to better infections outcomes [27,139,140]. Hence, the use of phages that target lipopolysaccharide (LPS) could lead to significant alterations to LPS produced by bacteria, and this might render them less virulent [141,142,143,144,145,146]. Thus, careful selection of the type of phage to be used in the fight against infectious diseases is important and should take into consideration the evolution of phage resistance to turn it into an advantage for humans and animals.

Currently, the application of artificial intelligence in the field of phage therapeutics is a trending topic since it can facilitate the selection of phages depending on the specific characteristics of the target pathogens and the host’s profile as well [147,148,149]. Hence, machine learning, which has been implemented broadly in biology, was shown to be able to integrate enormous amounts of information from omics data to better understand the phage–host interaction. This could be used for better identification of candidate phages for human medical use [149].

### 4.4. Efficacy of Phage Treatment in Animal Models

Phages have been found to be effective in the management of systemic infections in several animal models. In a gut-derived model of *Pseudomonas* sepsis, 67% survival was noted after oral administration of phage therapy one day after the infection of mice [150]. Phage dose-dependent reduction of mortality was noted in a mouse model of bacteremia by *Enterococcus faecium* resistant to vancomycin [151]. In another mouse model of infection by *Vibrio vulnificus*, successful treatment was noted only when the administration of phage therapy was performed at the same time as the infection by the bacteria [152]. Thus, the efficacy of phage treatment seems to rely on several factors that should be taken into account, such as the dose and the time after infection.

The use of phages for the treatment of localized infections such as an abscess, an ear infection, or a burn has been proven very efficient. An intraperitoneal model of infection by *P. aeruginosa* in mice treated at the same time with phages showed 92% survival [150]. In another study using *S. aureus* leading to the formation of abscesses, administration of phages at the same time as the pathogen prevented the formation of abscesses; meanwhile, when phage treatment was given 4 days after the bacterium inoculation, a single dose of phage treatment led to 100-fold reduction of the bacterial load, and when multiple doses of phage treatment were applied, 10,000-fold reduction was observed [153].

The use of phages against pathogens infecting the gastrointestinal tract may eradicate the pathogenic bacterium without altering the normal gut flora. Phage treatment four days after inoculation of adherent-invasive *E. coli* to the gut of mice led to reduced bacterial colonization and was associated with a reduced likelihood of colitis development [154]. A study that used an insect model of *Clostridioides difficile* colonization showed that prophylactic treatment with a phage two hours before inoculation of bacteria led to 100% survival. Simultaneous administration of phage and bacteria led to a 72% survival, while phage treatment two hours after inoculation of bacteria led to 30% survival [155].

Treatment of lung infections with phage therapy could help people with chronic lung infections, such as those with cystic fibrosis, who are also at particular risk of colonization and developing infections by antibiotic-resistant microorganisms. An experiment in mice using *P. aeruginosa* showed that intranasal treatment with two doses of phages led to complete eradication of the bacterium when administered 24/36 or 48/60 h after initiation of the infection, while treatment at 144/156 h after an infection led to complete eradication of the infection in 70% of the animals and a significant reduction of the bacterial load in the lungs of the rest [156].

Phage therapy may also have application in veterinary medicine, with phages being used against a variety of pathogens and in a variety of hosts. Hence, phages have been effectively tested against *Campylobacter jejuni* in chicken [157,158,159,160,161], against *Salmonella enterica* serovar Enteritidis in chicken [162,163,164], against *S. enterica* serovar Typhimurium in weaned pigs [165,166], against *S. aureus* in bovine mastitis [167,168], and against *S. aureus* in lactating dairy cattle, among others [169,170].

### 4.5. Combination of Phages with Antibiotics

There are several studies evaluating the effect of phage treatment on specific infections. However, there are few studies evaluating the effect of the combination treatment of phages with antibiotics, with some of them being promising. Hence, in a study of broiler chickens infected with *E. coli*, treatment with fluoroquinolone (enrofloxacin) led to a reduction in mortality from 68% down to 3%, while treatment with phages led to a reduction in mortality to 15%. Combination treatment with fluoroquinolone and phage simultaneously led to 0% mortality [171]. Another study showed that in a rat model of endocarditis induced by *P. aeruginosa*, combination treatment of ciprofloxacin and phage led to a 10,000-fold reduction of bacterial load compared to treatment with ciprofloxacin or phage alone; meanwhile, the same combination showed synergy in the killing of *P. aeruginosa* both in vitro and in vivo [172]. Thus, even though the topic of phage treatment is promising, studies evaluating both the in vitro and in vivo effects of phages with antibiotic combination treatment are warranted.

### 4.6. Studies in Humans

The use of phages in humans for the treatment of infectious diseases in the modern era is limited. However, there are studies evaluating their efficacy and safety [27,173]. There are reports of individual patients with limited therapeutic options treated with phages due to resistance or allergies to antibiotics. A two-year-old patient with DiGeorge syndrome and allergies to several antibiotics who developed *P. aeruginosa* bacteremia and failed treatment with anti-pseudomonal antimicrobial agents was treated with two-phage combination with temporary blood sterilization. However, after the end of this treatment, the blood cultures became positive again [174]. It is of note, though, that in this case, the recurrence of bacteremia does not necessarily imply that the phage treatment failed since bacteremia developed in the context of persistent and uncontrolled infected fluid collections of the thoracic cavity. Thus, phage treatment of uncomplicated bacteremia could be possible; however, this has yet to be confirmed by future studies.

In another case report, a patient with a *P.-aeruginosa*-infected aortic graft, complicated by aorto-cutaneous fistula with purulent discharge, failed very prolonged treatment with several antibiotics. Since, at that point in time, the patient was not a candidate for a new surgery, a phage active against *P. aeruginosa* that was shown to have synergy with ceftazidime and was screened for lytic activity against the causative organism was applied locally in the exit point of the fistula, along with systematic administration of ceftazidime [139,175]. One month after phage treatment, partial graft excision and replacement took place, while all cultures were negative, and systematic ceftazidime treatment was stopped. Two years later, the infection had not relapsed in the absence of antimicrobial treatment.

Recent clinical trials in humans often show contradictory findings [173]. However, there are studies with positive results that increase optimism that carefully and thoughtfully selected phages for the right pathogen and the right route of administration could be of clinical use in the future. A phase-one, first-in-humans, open-label clinical trial of multiple ascending doses was performed in a tertiary referral center in nine patients with recalcitrant chronic rhinosinusitis using the investigational phage cocktail AB-SA01. Intranasal treatment with AB-SA01 in different doses of up to 3 × 10^9^ PFU for two weeks was shown to be a safe and well-tolerated treatment, while the preliminary efficacy observations were promising [176]. This study, even though a phase-one study in a small number of patients, implies that in the future, treatment with phages could be an alternative to classic antibiotics in patients with recalcitrant chronic rhinosinusitis.

The PhagoBurn trial was a randomized phase I/II trial where adult patients with confirmed burn infection by *P. aeruginosa* were recruited in French and Belgian burn centers and received treatment after randomization with either standard treatment involving 1% sulfadiazine silver emulsion cream or a cocktail of 12 natural lytic anti-*P.-aeruginosa* phages, and the time to an adequate sustained reduction of bacterial burden was assessed. The phage cocktail was found to decrease the bacterial burden in burn wounds at a slower pace compared to the standard of care, implying that future studies with higher concentrations of phages are required [177]. Table 1 shows the studies of phage treatment that have been conducted in humans.

### 4.7. Authorization by Regulatory Authorities

Nowadays, FDA approval has not yet been granted to phage treatment, and this could be due to concerns regarding approving a medication that is “alive” and difficult to standardize. Thus, since concerns still exist and hinder the approval of this modality, other approaches such as developing recombinant phage-derived proteins could be elaborated [188,189]. Even though this treatment is not yet approved, there are some clinical trials that have been registered and are currently active [190,191]. Clinicians in the USA who intend to use phages in patients must submit an investigational new drug application to the FDA [192]. Similarly, the European Medicines Agency (EMA) has not yet approved any bacteriophage for clinical use [193], though the draft guidelines on “quality, safety and efficacy of bacteriophages as veterinary medicines” are now undergoing an expert public consultation [194]. Moreover, a chapter on bacteriophage use in therapy will shortly be introduced in the European Pharmacopoeia [195]. There are several barriers worldwide to the production and application of phages as an alternative or a complementary therapy to classical antimicrobials. The most important one is the lack of adequate data from clinical trials set up based on widely accepted ethical standards. There are, however, countries such as Poland, Georgia, and Russia where phage therapy is used, even though there are no clearly adapted regulatory guidelines [196]. In Poland, however, treatment with phages is considered “experimental” under the Polish Law Gazette, 2011, item 1634 and article 37 of the Declaration of Helsinki [193,197].

## 5. Conclusions

Antimicrobial resistance has emerged as an enormous problem threatening millions of lives and causing thousands of infections in patients, most commonly in hospitalized ones. Since many pathogens have multiple mechanisms of resistance and have few viable therapeutic options, alternative treatments for infectious diseases are warranted. To that end, the use of phages, which were discovered at the beginning of the previous century and have shown some clinical success in early trials at that time, could now improve the treatment of patients suffering infections from difficult-to-treat pathogens. Until now, there has been plenty of evidence from animal models about the safety and efficacy of phage therapy, and phages are used broadly in agriculture, aquaculture, food safety, and wastewater plant treatment and as hospital environment sanitizers. However, there is a lack of high-quality evidence regarding phage therapy in clinical practice, with most studies being case reports or non-controlled trials often with contradictory results. Further well-designed studies with randomized, blind, controlled designs and with careful and thoughtful selection of phages and respect to global standards set by regulatory authorities are needed to identify the role of this promising treatment in the future of infectious diseases. Until then, phage use in clinical practice remains non-approved by the FDA and the EMA.

## Figures and Tables

**Figure 1 antibiotics-12-01012-f001:**
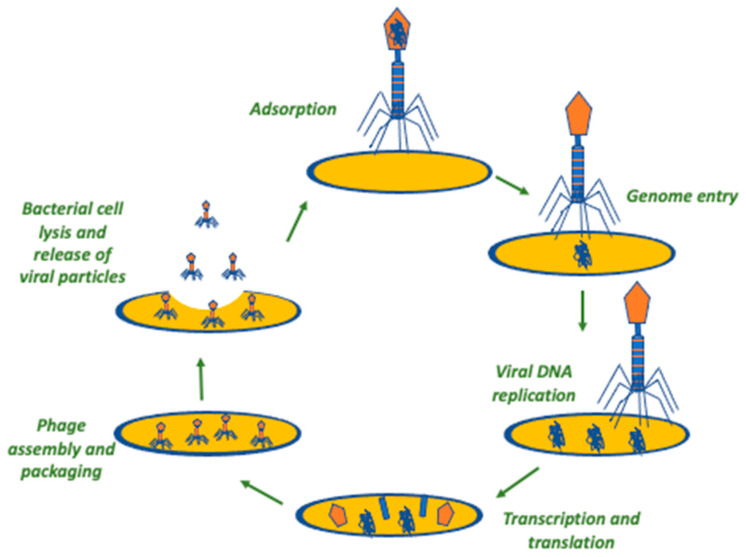
The lytic phage cycle. In this procedure, the irreversible binding of phage to the surface of a bacterial cell is the first step. Secondly, the phage transfers its genome into the cytoplasm of the bacterial cell, proceeding to its replication. Then, the metabolism of the bacterial cell is leveraged for the assembly of newly replicated phages that, in turn, are released from the bacterial cell through lysis.

**Figure 2 antibiotics-12-01012-f002:**
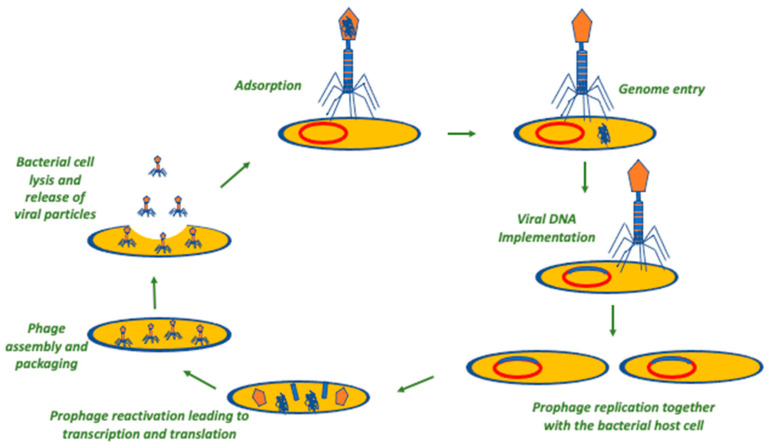
The temperate/lysogenic phage cycle. In this procedure, the irreversible binding of phage to the surface of a bacterial cell is the first step. Secondly, the phage transfers its genome into the cytoplasm of the bacterial cell, and this is inserted into the bacterial genome. This may remain in a dormant stage (prophage). In the appropriate environmental conditions, the prophage is activated, leading to the reactivation of the lytic lifecycle.

**Figure 3 antibiotics-12-01012-f003:**
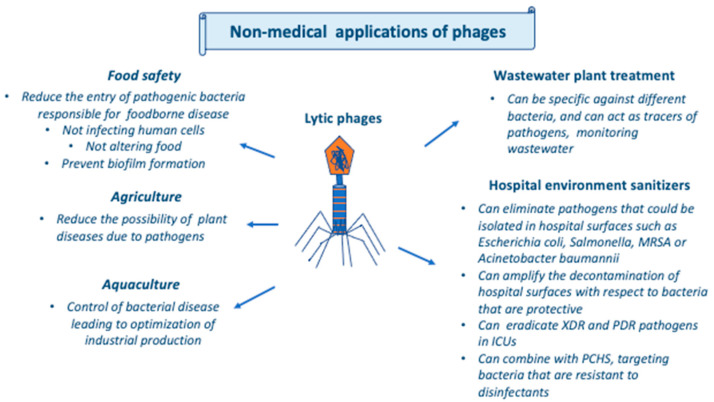
Non-medical applications of phages. Abbreviations: ICU, intensive care unit; MRSA, methicillin-resistant *Staphylococcus aureus*; PCHS, probiotic cleaning hygiene system; PDR, pan-drug-resistant; XDR, extensively drug-resistant.

**Table 1 antibiotics-12-01012-t001:** Studies evaluating clinical use of phages in humans.

Study	Population	Intervention	Comparator	Outcome
Jault et al., 2019 [177] (PhagoBurn)—Phase I/II trial	Adult patients with burns infected by *P. aeruginosa*	Cocktail by 12 anti-*P.-aeruginosa* phages (PP1131)	Standard of care (1% sulfadiazine silver emulsion cream)	Phage cocktail reduced bacterial burden more slowly than the standard of care
Ooi et al., 2019 [150]—Phase I trial	Nine patients with recalcitrant chronic rhinosinusitis (18–70 years old) with failure of surgical and medical treatment and positive cultures for *S. aureus* sensitive to investigational phage cocktail AB-SA01	Serial doses of twice-daily intranasal irrigations with AB-SA01	None	Intranasal irrigation with AB-SA01 was safe and well tolerated
Wright et al., 2009 [178] —Phase I/II trial	24 patients with chronic otitis with positive culture for antibiotic-resistant *P. aeruginosa* sensitive to Biophage-PA	A single dose of Biophage-PA (10^9^ directly in the ear) after randomization	Placebo	Poled patient- and physician-reported clinical indicators improved for the phage-treated group relative to the placebo group. No treatment-related adverse event was reported
Sarker et al., 2016 [179]—Double-blind, placebo-controlled	Bangladeshi children hospitalized with acute bacterial diarrhea	40 individuals received phage cocktail M, and 39 individuals received phage cocktail T orally three times daily in oral rehydration solution over 4 days	Placebo (oral rehydration solution)	No significant difference between the group treated with phages and the placebo group was noted
Leitner et al., 2021 [180]—Randomized, placebo-controlled trial	Adult males scheduled for TURP, with complicated UTI or recurrent uncomplicated UTIs	28 patients received at least one intravesical dose of Pyophage after randomization (the planned dose was twice daily for seven days)	32 patients received a placebo and received 37 systematic antibiotics after randomization	Intravesical bacteriophage therapy was non-inferior to standard-of-care antibiotic treatment but was not superior to placebo bladder irrigation in terms of efficacy or safety
Rhoads et al., 2009 [181]—Phase I trial	42 patients with chronic venous leg ulcers	The ulcers were treated for 12 weeks with bacteriophages targeted against *P. aeruginosa*, *S. aureus*, and *E. coli*	Saline control	No adverse events due to phages. No significant difference for frequency of adverse events, rate of healing, or in the frequency of healing
Samaee et al., 2023 [182]—Double-blind, placebo-controlled, randomized study	60 patients with moderate-to-severe COVID-19	For the intervention group, 10 mL of phage cocktail with a titer of 10^12^ PFU/mL was given with a mesh nebulizer	The control group received 10 mL of phage-free suspension (placebo) every 12 h with a mesh nebulizer	Inhalation phage therapy may have a potential effect on secondary infection and on the outcome of COVID-19 patients
Fedorov et al., 2023 [183]—Non-randomized, open-label, with historical control study	Adult patients with deep PJI of the hip with a 12-month follow-up after one-stage revision surgery	23 patients were treated with specific phage preparation and etiotropic antibiotics	22 patients from a retrospective historical control group received antibiotics only	PJI relapses in the intervention group were eight times lower. The response rate to treatment was 95.5% in the intervention and only 63.6% in the control
Petrovic Fabijan et al., 2020 [184]—Single-arm, non-comparative trial	Adult patients with two consecutive days of *S. aureus* bacteremia	13 patients were administered adjunctive AB-SA01 intravenously	None	No adverse reactions were reported, and AB-SA01 appeared to be safe in severe *S. aureus* infections, including septic shock and infective endocarditis
Duplessis et al., 2018 [174]—Case report	Two-year-old boy with DiGeorge syndrome and congenital heart disease and pacemaker placement with *P. aeruginosa* bacteremia	Bacteriophage cocktail active against that specific *P. aeruginosa* isolate	None	Blood cultures sterile after treatment
Chan et al., [175]—Case report	A 76-year-old patient with infected aortic graft due to *P. aeruginosa* and complicated by aorto-cutaneous fistula with purulent discharge	A phage active against *P. aeruginosa* that had synergy with ceftazidime was applied locally in the exit point of the fistula, along with systematic administration of ceftazidime. Partial graft excision and replacement took place	None	Cultures were sterile one month later. Two years later, the infection had not relapsed in the absence of antimicrobial treatment
Khawaldeh et al., 2011 [185]—Case report	A 67-year-old woman with extensive intra-abdominal resections and pelvic irradiation for adenocarcinoma, bilateral ureteric stent placement for obstruction complicated by *P. aeruginosa* infection, and with multiple courses of antibiotics and two stent replacements	2 × 10^7^ PFU of a lytic phage active against the infecting strain was directly instilled into the bladder every 12 h for 10 days (antibiotics also started on day 6)	None	Urine cultures were sterile after phage therapy and a 30-day course of meropenem
LaVergne et al., 2018 [186]—Case report	A 77-year-old man with traumatic brain injury who underwent craniectomy and was complicated by postoperative infection by XDR *A. baumannii*	8.56 × 10^7^ PFU of active phage for that bacterial strain administered intravenously every 2 h for 8 days	None	Initial patient improvement was observed, and craniotomy site and skin flap healed well, but fevers and leukocytosis persisted. The patient died after care withdrawal
Schooley et al., 2017 [187]—Case report	68-year-old diabetic man with necrotizing pancreatitis complicated by an MDR *A.-baumannii*-infected pseudocyst	5 × 10^9^ PFU administered intravenously every 6 h for 84 days, with minocycline being added on day two	None	The patient improved clinically and the infection resolved

MDR, multi-drug-resistant; PJI, prosthetic joint infection; TURP, transurethral resection of the prostate; UTI, urinary tract infection; XDR, extensively drug-resistant. This table may not be exhaustive of all studies in the literature.

## Data Availability

The data presented in this study are available on request from the corresponding authors.

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
