# Peer review of "Bacteriophages in Infectious Diseases and Beyond—A Narrative Review"

_antibiotics, 2023, doi:10.3390/antibiotics12061012_

Round 1

Reviewer 1 Report

The manuscript addresses an actual topic as the emerging resistance to antibiotics is a global concern, and the development of novel therapeutic approaches is gaining attention. However, some important points are missing and should be discussed by the authors:

1. The authors correctly suggest how phages may be used in clinical practice, but an important aspect that prevents this development is what concerns FDA approval. Phages are "live" therapeutics that are quite impossible to standardize, thus hindering the approval process. A possible solution might be the development of recombinant phage-derived proteins, the authors should mention at least this possibility (for example doi: 10.1042/BST20150192; 10.1155/2017/3780697).

2. The main limit of phage therapy is the extremely reduced host spectrum of a single bacteriophage. Thus, Lines 215-220 are incorrect: it is extremely unlikely the off-target of a phage, so the fact that "this may mean that beneficial bacteria can be targeted as well" cannot happen with a well-known preparation of selected phages. Also, Lines 294-297 and 300-301 contain a conceptual error, saying "an important drawback at that time was the phages’ high specificity against certain bacteria" and "This inconsistency was considered due to using a phage with very narrow spectrum" because it is due to the physiology of any bacteriophage structure and life cycle. To this aim, the work by Dedrick et al. (doi: 10.1038/s41591-019-0437-z) described how a collection of >10,000 phages isolated using Mycobacterium smegmatis needed to be screened to identify just three useful phages, two of which to be engineered because they were too ineffective. Please add this concept in the introduction, and this example and some others humans application to topic infections (i.e. PhagoBurn clinical trial) to the 4.6. Moreover, a table summarizing all studies in humans of phage therapy could be helpful for the readers. Finally, the most recent studies are trying to exploit AI and specific algorithms to predict the more effective phage-therapy cocktails considering the available collections of phages and their characteristics of specificity, patient's factors, and pathogens factors as well (doi: 10.1038/s41598-022-06422-1).

3. Line 48: The concept that "antibiotic combinations" limit the spreading of antibiotic resistance should be explicated, since it's not as straightforward as the concept of using old molecules.

4. Line 96: The molecular interactions between phage proteins and host receptors are fundamental, as already highlighted in point 2. Please move here the part describing the examples of these interactions (Lines 118-132), expanding it.

5. Food safety part and medical applications: bacteriophages are used also as biosensors in both quality control procedures and diagnosis. Please add a few phrases to both sections.

6. Part 3.5 should describe also the biofilm treatment of surfaces and medical equipment, not only the environmental microbiome as free bacterial species colonizing the hospital places. Moreover, authors should explicate that phages added to probiotic systems don't kill the main component/s of PCHS itself (Line 253).

7. Please better address the concept of bacterial resistance to phages because it is only mentioned in a non-exhaustive way.

Minor points:
- Figures 1 and 2: please, use a darker shade for the writings because they are barely seen. The authors may add the lysogenic cycle to Figure 1 to complete the phage cycle description, or add another figure to address this point.

- Line 15 "to treat resistant to antibiotics pathogens" is unclear, please correct

- Line 71 I think the authors intended "diseases", "disease"

- Line 112 Please specify better "specific environmental or physiological triggering", as it is a bit vague. It is known that depends mainly on the fitness of the host cell.

The English language must be revised to improve the manuscript's clarity, in particular for the Abstract and the Introduction sections.

Author Response

The manuscript addresses an actual topic as the emerging resistance to antibiotics is a global concern, and the development of novel therapeutic approaches is gaining attention. However, some important points are missing and should be discussed by the authors:

  1. The authors correctly suggest how phages may be used in clinical practice, but an important aspect that prevents this development is what concerns FDA approval. Phages are "live" therapeutics that are quite impossible to standardize, thus hindering the approval process. A possible solution might be the development of recombinant phage-derived proteins, the authors should mention at least this possibility (for example doi: 10.1042/BST20150192; 10.1155/2017/3780697).

Response: Thanks for the comment. Indeed, there are still concerns regarding granting authorization from FDA to phage treatment, and developing new approaches using proteins from phages as antimicrobial peptides could be a solution that could bypass the difficulty of using a live therapeutic modality. Thus, we added this concern and the corresponding references after the subsection 4.6, the newly written 4.7 subsection (per another reviewer’s request) as can be seen in the revised version of the manuscript, as suggested by the reviewer.

  1. The main limit of phage therapy is the extremely reduced host spectrum of a single bacteriophage. Thus, Lines 215-220 are incorrect: it is extremely unlikely the off-target of a phage, so the fact that "this may mean that beneficial bacteria can be targeted as well" cannot happen with a well-known preparation of selected phages. Also, Lines 294-297 and 300-301 contain a conceptual error, saying "an important drawback at that time was the phages’ high specificity against certain bacteria" and "This inconsistency was considered due to using a phage with very narrow spectrum" because it is due to the physiology of any bacteriophage structure and life cycle. To this aim, the work by Dedrick et al. (doi: 10.1038/s41591-019-0437-z) described how a collection of >10,000 phages isolated using Mycobacterium smegmatis needed to be screened to identify just three useful phages, two of which to be engineered because they were too ineffective. Please add this concept in the introduction, and this example and some others humans application to topic infections (i.e. PhagoBurn clinical trial) to the 4.6. Moreover, a table summarizing all studies in humans of phage therapy could be helpful for the readers. Finally, the most recent studies are trying to exploit AI and specific algorithms to predict the more effective phage-therapy cocktails considering the available collections of phages and their characteristics of specificity, patient's factors, and pathogens factors as well (doi: 10.1038/s41598-022-06422-1).

Response: This is a useful comment. We deleted that sentence in lines 215-220, and now we do not imply that phages could target beneficial bacteria in wastewater plants, even when polyvalent mixtures are used. Regarding the narrow spectrum of a phage, we do agree in principle that phages have very narrow targets; however, it is possible to expand their targets through phage engineering (10.1128/mbio.00472-23, 10.1002/btm2.10381). This makes our statement correct that in the previous decades the spectrum of phages was narrower than it could be today. We are sorry we hadn’t mentioned phage engineering in the manuscript before. We have now changed the manuscript to include such information to allow the reader to understand that the basic physiological properties of phages that render them unable to target multiple bacterial targets can be somewhat manipulated today. We did not change the introduction section, since we hadn’t mentioned phages at that part, but we did change section 2 right at the beginning, as can be seen in the revised version of the manuscript, to allow the reader to understand how specific phages are. Since, during the revisions, we mentioned the study of Dedrick et al. twice (once in section 2 and once in 4.3), we chose not to mention it again in 4.6. However, we did mention the PhagoBurn trial, and we created a table mentioning the vast majority of the studies evaluating phage therapy in humans. However, this table may not be exhaustive. Finally, we added some information about the possible applications of artificial intelligence in the medical use of phages.

  1. Line 48: The concept that "antibiotic combinations" limit the spreading of antibiotic resistance should be explicated, since it's not as straightforward as the concept of using old molecules.

Response: Thanks. We have added a sentence right after the sentence mentioned by the reviewer to allow the reader to understand the concept of antimicrobial combinations and their use in PDR microorganisms such as in A. baumannii. This can be seen in the introduction of the revised version of our manuscript.

  1. Line 96: The molecular interactions between phage proteins and host receptors are fundamental, as already highlighted in point 2. Please move here the part describing the examples of these interactions (Lines 118-132), expanding it.

Response: Thanks. We have moved that part at the designated point by the reviewer. Furthermore, we expanded that part by adding some examples of interactions between phages and their target receptors. This can be seen in the corresponding part in the revised version of the manuscript.

  1. Food safety part and medical applications: bacteriophages are used also as biosensors in both quality control procedures and diagnosis. Please add a few phrases to both sections.

Response: We added a paragraph in the subsection discussing the applications of phages in food safety and the sentence in the subsection discussing the use of phages in medical applications (not for use in humans). There, we mention the possible application of phages in the context of biosensors and we also added some representative references that discuss the topic more thoroughly. This can be seen in the corresponding parts in the revised version of the manuscript.

  1. Part 3.5 should describe also the biofilm treatment of surfaces and medical equipment, not only the environmental microbiome as free bacterial species colonizing the hospital places. Moreover, authors should explicate that phages added to probiotic systems don't kill the main component/s of PCHS itself (Line 253).

Response: Useful comment. We added another paragraph at the end of 3.5 subsection to specifically mention the effect of phage treatment in the biofilm in the hospital environment. Furthermore, we specifically mentioned that phages added to the PCHS do not kill the non-pathogenic probiotic bacteria they contain. These changes can be seen in 3.5 subsection of the revised version of the manuscript.

  1. Please better address the concept of bacterial resistance to phages because it is only mentioned in a non-exhaustive way.

Response: Thanks. We have expanded on the topic of development of bacterial resistance to phage treatment in the subsection 4.1 of the revised version of the manuscript. We feel that the reader can now understand the mechanism of resistance development as well as its adaptive nature. We have also added some references on the topic.

Minor points:
- Figures 1 and 2: please, use a darker shade for the writings because they are barely seen. The authors may add the lysogenic cycle to Figure 1 to complete the phage cycle description, or add another figure to address this point.

Response: We made the requested changes. The figures and the characters are now darker and we created a new figure to show the cycle of a lysogenic phage.

- Line 15 "to treat resistant to antibiotics pathogens" is unclear, please correct

Response: We corrected that as can be seen in the abstract of the revised version of the manuscripts.

- Line 71 I think the authors intended "diseases", "disease"

Response: That is correct. We changed that.

- Line 112 Please specify better "specific environmental or physiological triggering", as it is a bit vague. It is known that depends mainly on the fitness of the host cell.

Response: We changed that to ‘upon specific environmental or physiological stressor that negatively affects the fitness of the bacterial host cell’ to make it clearer for the reader to understand that; indeed, the activation of the prophage mostly depends on the fitness of the host bacterial cell.

Comments on the Quality of English Language

The English language must be revised to improve the manuscript's clarity, in particular for the Abstract and the Introduction sections.

Response: Thanks. The manuscript was revised by a native English speaker and modifications were done throughout the text. We feel that the revised manuscript is now improved.

Reviewer 2 Report

For giving a clearer structure to this draft manuscript, to my personal point of view, chapt 3 “Non-medical applications of phages” should be placed after “chapt on medical application”. Moreover, this chapter about “non –medical applications of phages” must be enriched with more details  about the results of the experiments. Those experiments are only very briefly reported without any reference to the results activities of those bacteriophage applications. Please, check this also all over the text.

Please, it is up to the authors to decide if it is or not the case to add a paragraph of “in vitro studies of phage efficacy”. But this reviewer thinks that it is necessary to add a paragraph (or integrate the one with mouse model) with the “In vivo studies on animal target”. Please, address the most common animal diseases or zoonoses up to date and those experiments already carried out in the field also according to the last report on zoonoses (European Food Safety Authority (EFSA), Amore, G., Boelaert, F., Gibin, D., Papanikolaou, A., Rizzi, V., & Stoicescu, A. V. (2022). Zoonoses and foodborne outbreaks guidance for reporting 2021 data (Vol. 19, No. 1, p. 7131E). Please, among the others, take in consideration e.g. phage therapy against Campylobacter in poultry, being this one of the most common zoonoses in humans. Suggest to add the following ref: “D’angelantonio, D., Scattolini, S., Boni, A., Neri, D., Di Serafino, G., Connerton, P., ... & Aprea, G. (2021). Bacteriophage therapy to reduce colonization of campylobacter jejuni in broiler chickens before slaughter. Viruses, 13(8), 1428.”

A Chapter on the Legal framework about phage applications in the world is also needed. Please, report the current status about EMA (https://www.ema.europa.eu/en/quality-safety-efficacy-bacteriophages-veterinary-medicines-scientific-guideline) and European Pharmacopoeia (https://www.edqm.eu/en/about-edqm/-/asset_publisher/wQwK2Umbt4vx/content/public-consultation-on-new-general-chapter-on-phage-therapy-active-substances-and-medicinal-products-for-human-and-veterinary-use-in-pharmeuropa-35.2?_com_liferay_asset_publisher_web_portlet_AssetPublisherPortlet_INSTANCE_wQwK2Umbt4vx_assetEntryId=1630069&_com_liferay_asset_publisher_web_portlet_AssetPublisherPortlet_INSTANCE_wQwK2Umbt4vx_redirect=https%3A%2F%2Fwww.edqm.eu%2Fen%2Fabout-edqm%3Fp_p_id%3Dcom_liferay_asset_publisher_web_portlet_AssetPublisherPortlet_INSTANCE_wQwK2Umbt4vx%26p_p_lifecycle%3D0%26p_p_state%3Dnormal%26p_p_mode%3Dview%26_com_liferay_asset_publisher_web_portlet_AssetPublisherPortlet_INSTANCE_wQwK2Umbt4vx_cur%3D0%26p_r_p_resetCur%3Dfalse%26_com_liferay_asset_publisher_web_portlet_AssetPublisherPortlet_INSTANCE_wQwK2Umbt4vx_assetEntryId%3D1630069).

Conclusions should be rewritten and more developed.

More comments are reported in the revised copy of the manuscript that has been uploaded.

English grammar must be extensively revised by a native EN speaker all over the text.

Author Response

For giving a clearer structure to this draft manuscript, to my personal point of view, chapt 3 “Non-medical applications of phages” should be placed after “chapt on medical application”. Moreover, this chapter about “non –medical applications of phages” must be enriched with more details  about the results of the experiments. Those experiments are only very briefly reported without any reference to the results activities of those bacteriophage applications. Please, check this also all over the text.

Response: Thanks for the comment. To be honest, when drafting this manuscript, the basic aim was to focus on the medical applications, either clinical use in humans, or non-human medical use within the hospital environment (such as for disinfection of hospital surfaces). The other applications were added just to give the reader an idea of the multiple uses phages have in topics unrelated to medicine. Since our main topic is still the medical use of phages, we don’t agree with the reviewer, and we would prefer to keep the section of non-medical application of phages before the medical uses, since this would allow the reader to take a brief idea about the phage biology and applications, before moving on to the medical applications and the clinical use. We feel that it is also evident in the revised version of the present manuscript that our main focus was the medical use of phages. For example, a new table was added, mentioning almost all the studies involving phage use in medicine. We did, however, briefly expand some parts of the non-medical applications of phages, as requested by the reviewer. However, we did not do that extensively, since the size of the manuscript is already very big (13,866 words and 175 references until the moment we address this comment).

Please, it is up to the authors to decide if it is or not the case to add a paragraph of “in vitro studies of phage efficacy”. But this reviewer thinks that it is necessary to add a paragraph (or integrate the one with mouse model) with the “In vivo studies on animal target”. Please, address the most common animal diseases or zoonoses up to date and those experiments already carried out in the field also according to the last report on zoonoses (European Food Safety Authority (EFSA), Amore, G., Boelaert, F., Gibin, D., Papanikolaou, A., Rizzi, V., & Stoicescu, A. V. (2022). Zoonoses and foodborne outbreaks guidance for reporting 2021 data (Vol. 19, No. 1, p. 7131E). Please, among the others, take in consideration e.g. phage therapy against Campylobacter in poultry, being this one of the most common zoonoses in humans. Suggest to add the following ref: “D’angelantonio, D., Scattolini, S., Boni, A., Neri, D., Di Serafino, G., Connerton, P., ... & Aprea, G. (2021). Bacteriophage therapy to reduce colonization of campylobacter jejuni in broiler chickens before slaughter. Viruses, 13(8), 1428.”

Response: Thanks for the comment. We added a paragraph at the end of 4.4 mentioning the use of phage therapy in veterinary medicine, with many examples including therapy of Campylobacter in chicken, or Salmonella in chicken and pigs. However, we did not expand that paragraph that much, since we feel that the manuscript is already very extended.

A Chapter on the Legal framework about phage applications in the world is also needed. Please, report the current status about EMA (https://www.ema.europa.eu/en/quality-safety-efficacy-bacteriophages-veterinary-medicines-scientific-guideline) and European Pharmacopoeia (https://www.edqm.eu/en/about-edqm/-/asset_publisher/wQwK2Umbt4vx/content/public-consultation-on-new-general-chapter-on-phage-therapy-active-substances-and-medicinal-products-for-human-and-veterinary-use-in-pharmeuropa-35.2?_com_liferay_asset_publisher_web_portlet_AssetPublisherPortlet_INSTANCE_wQwK2Umbt4vx_assetEntryId=1630069&_com_liferay_asset_publisher_web_portlet_AssetPublisherPortlet_INSTANCE_wQwK2Umbt4vx_redirect=https%3A%2F%2Fwww.edqm.eu%2Fen%2Fabout-edqm%3Fp_p_id%3Dcom_liferay_asset_publisher_web_portlet_AssetPublisherPortlet_INSTANCE_wQwK2Umbt4vx%26p_p_lifecycle%3D0%26p_p_state%3Dnormal%26p_p_mode%3Dview%26_com_liferay_asset_publisher_web_portlet_AssetPublisherPortlet_INSTANCE_wQwK2Umbt4vx_cur%3D0%26p_r_p_resetCur%3Dfalse%26_com_liferay_asset_publisher_web_portlet_AssetPublisherPortlet_INSTANCE_wQwK2Umbt4vx_assetEntryId%3D1630069).

Response: Thanks for the comment. We added a subsection, 4.7, that discusses the status of authorization of phage for clinical use by the regulatory authorities, namely FDA and EMA. Actually, even though there are some studies that have been accepted to enrol patients by the FDA, and these protocols have been published, there is still no approved phage therapy by either the FDA or the EMA. There are specific concerns regarding their use, and these concerns mainly have to do with the fact that they are actually a living therapy, while, there is a relative lack of data from well-designed trials worldwide. This information now can be found before the conclusions section of the revised version of the manuscript.

Conclusions should be rewritten and more developed.

Response: We have changed the conclusions section by adding more information to it, mentioning the non-medical use of phages, and the need for sophisticated future clinical trials that could lead to the approval of this treatment by the regulatory authorities. This can be seen at the conclusions section of the revised version of the manuscript.

More comments are reported in the revised copy of the manuscript that has been uploaded.

Response: Thanks for the comments in the pdf. We have carefully checked that and made all the suggested modifications in the text.

Comments on the Quality of English Language

English grammar must be extensively revised by a native EN speaker all over the text.

Response: The manuscript was revised by a native English speaker and modifications were done throughout the text. We feel that the revised manuscript is now improved.

Reviewer 3 Report

1.The manuscript title is "Bacteriaphages in treatment of infectous disease",howerer,phages application in food safety,in agriculture,in aquaculture ,et al had been reviewed,I think the title is not suitable.

2."Bacteriaphage""phage",both of  words were included in keyword,one of them may delete. 

Author Response

1.The manuscript title is "Bacteriaphages in treatment of infectous disease",howerer,phages application in food safety,in agriculture,in aquaculture ,et al had been reviewed,I think the title is not suitable.

Response: Thanks for the comment. We changed the title to ‘Bacteriophages in Infectious Diseases and beyond – a narrative review’. We feel that it better reflects the content of the manuscript now.

2."Bacteriaphage""phage",both of  words were included in keyword,one of them may delete.

Response: Thanks. We deleted the keyword ‘phage’.

Round 2

Reviewer 1 Report

The authors addressed all my comments and the manuscript results improved.

Author Response

The authors addressed all my comments and the manuscript results improved.

Response: Thanks for the nice comments.

Reviewer 2 Report

Most of the comments have been addressed by the authors.

Still few minor revisions are needed:

Line 51: A. baumannii. Please, report the Genus of the bacteria in the extensive way the first time it is cited in the text. Apply to all the bacteria for the first time they appear in the text eg. Acinetobacter baumannii (A. baumannii)

Line 58: with the potential TO BE TESTED IN clinical studies

Line 107: The lytic phage cycle.

Line 113: Figure 2. The temperate/lysogenic phage cycle

Line 136: …to the bacterial receptor AND THE GENOME INJECTION INTO THE CELL, the genetic material…..

Line 152: …stressorS …..

Line 158: ……the life cycleS….

Line 231: TO eliminate

Line 308: due to their specificity for bacteria acting

Lines 378 – 381: repetitions to be avoided

Lines 421 – 423: repetitions to be avoided

Lines 484: interaction: repetitions to be avoided

Line 618: FDA acronym for?

Lines 618, 622, 624: FDA approval: repetitions to be avoided

Line 627: …for clinical use (193), though the draft guidelines on “quality, safety and efficacy of bacteriophages as veterinary medicines” are now undergoing an expert public consultation (https://www.ema.europa.eu/en/quality-safety-efficacy-bacteriophages-veterinary-medicines-scientific-guideline). Also a chapter on bacteriophage use in therapy is going to be introduced in the European Pharmacopoeia (https://www.edqm.eu/en/about-edqm/-/asset_publisher/wQwK2Umbt4vx/content/public-consultation-on-new-general-chapter-on-phage-therapy-active-substances-and-medicinal-products-for-human-and-veterinary-use-in-pharmeuropa-35.2?_com_liferay_asset_publisher_web_portlet_AssetPublisherPortlet_INSTANCE_wQwK2Umbt4vx_assetEntryId=1630069&_com_liferay_asset_publisher_web_portlet_AssetPublisherPortlet_INSTANCE_wQwK2Umbt4vx_redirect=https%3A%2F%2Fwww.edqm.eu%2Fen%2Fabout-edqm%3Fp_p_id%3Dcom_liferay_asset_publisher_web_portlet_AssetPublisherPortlet_INSTANCE_wQwK2Umbt4vx%26p_p_lifecycle%3D0%26p_p_state%3Dnormal%26p_p_mode%3Dview%26_com_liferay_asset_publisher_web_portlet_AssetPublisherPortlet_INSTANCE_wQwK2Umbt4vx_cur%3D0%26p_r_p_resetCur%3Dfalse%26_com_liferay_asset_publisher_web_portlet_AssetPublisherPortlet_INSTANCE_wQwK2Umbt4vx_assetEntryId%3D1630069).

Line 639: avoid the repetitions “treatment” , also present in line 642

Author Response

Most of the comments have been addressed by the authors.

Still few minor revisions are needed:

Line 51: A. baumannii. Please, report the Genus of the bacteria in the extensive way the first time it is cited in the text. Apply to all the bacteria for the first time they appear in the text eg. Acinetobacter baumannii (A. baumannii)

Response: Thanks. We changed that throughout the text.

Line 58: with the potential TO BE TESTED IN clinical studies

Response: We corrected that.

Line 107: The lytic phage cycle.

Response: We corrected that.

Line 113: Figure 2. The temperate/lysogenic phage cycle

Response: Thanks. We changed that.

Line 136: …to the bacterial receptor AND THE GENOME INJECTION INTO THE CELL, the genetic material…..

Response: That was corrected, as suggested by the reviewer.

Line 152: …stressorS …..

Response: We changed that.

Line 158: ……the life cycleS….

Response: That was corrected

Line 231: TO eliminate

Response: Thanks. We corrected that.

Line 308: due to their specificity for bacteria acting

Response: That was changed.

Lines 378 – 381: repetitions to be avoided

Response: Thanks. We changed that to reduce repetition.

Lines 421 – 423: repetitions to be avoided

Response: We changed that sentence to reduce repetition.

Lines 484: interaction: repetitions to be avoided

Response: We changed that sentence.

Line 618: FDA acronym for?

Response: It stands for Food and Drug Administration. It was first introduced at the point when it was first mentioned in page 5. That was in line 204 in the previous manuscript.

Lines 618, 622, 624: FDA approval: repetitions to be avoided

Response: Indeed, there was some repetition. We changed that paragraph to reduce it as can be seen in the revised version of the manuscript.

Line 627: …for clinical use (193), though the draft guidelines on “quality, safety and efficacy of bacteriophages as veterinary medicines” are now undergoing an expert public consultation (https://www.ema.europa.eu/en/quality-safety-efficacy-bacteriophages-veterinary-medicines-scientific-guideline). Also a chapter on bacteriophage use in therapy is going to be introduced in the European Pharmacopoeia (https://www.edqm.eu/en/about-edqm/-/asset_publisher/wQwK2Umbt4vx/content/public-consultation-on-new-general-chapter-on-phage-therapy-active-substances-and-medicinal-products-for-human-and-veterinary-use-in-pharmeuropa-35.2?_com_liferay_asset_publisher_web_portlet_AssetPublisherPortlet_INSTANCE_wQwK2Umbt4vx_assetEntryId=1630069&_com_liferay_asset_publisher_web_portlet_AssetPublisherPortlet_INSTANCE_wQwK2Umbt4vx_redirect=https%3A%2F%2Fwww.edqm.eu%2Fen%2Fabout-edqm%3Fp_p_id%3Dcom_liferay_asset_publisher_web_portlet_AssetPublisherPortlet_INSTANCE_wQwK2Umbt4vx%26p_p_lifecycle%3D0%26p_p_state%3Dnormal%26p_p_mode%3Dview%26_com_liferay_asset_publisher_web_portlet_AssetPublisherPortlet_INSTANCE_wQwK2Umbt4vx_cur%3D0%26p_r_p_resetCur%3Dfalse%26_com_liferay_asset_publisher_web_portlet_AssetPublisherPortlet_INSTANCE_wQwK2Umbt4vx_assetEntryId%3D1630069).

Response: Thanks for the comment. We changed that part by introducing the information and the references suggested by the reviewer.

Line 639: avoid the repetitions “treatment” , also present in line 642

Response: We changed the conclusions section by substituting some of those words with other synonyms to reduce repetition. This can be seen in the conclusions section of the revised manuscript.